# Characteristics of Bipolar Patients with Cognitive Impairment of Suspected Neurodegenerative Origin: A Multicenter Cohort

**DOI:** 10.3390/jpm11111183

**Published:** 2021-11-11

**Authors:** Esteban Munoz Musat, Emeline Marlinge, Mélanie Leroy, Emilie Olié, Eloi Magnin, Florence Lebert, Audrey Gabelle, Djamila Bennabi, Fréderic Blanc, Claire Paquet, Emmanuel Cognat

**Affiliations:** 1Cognitive Neurology Center, Assistance Publique—Hôpitaux de Paris Nord, Site Lariboisière Fernand-Widal, F-75010 Paris, France; estebanmunozmus@gmail.com (E.M.M.); claire.paquet@aphp.fr (C.P.); 2Department of Psychiatry and Addictive Medicine, Assistance Publique—Hôpitaux de Paris Nord, Site Lariboisière Fernand-Widal, F-75010 Paris, France; emeline.marlinge@aphp.fr; 3Lille Neuroscience & Cognition, Centre Hospitalier Régional Universitaire de Lille, Institut National de la Santé et de la Recherche Médicale, Univ. Lille, F-59000 Lille, France; melanie.leroy@chru-lille.fr; 4Centre National de Référence Maladie d’Alzheimer Jeune, Institut National de la Santé et de la Recherche Mé-dicale, Univ. Lille, F-59000 Lille, France; 5Lille Centre of Excellence for Neurodegenerative Disorders (LiCEND), Institut National de la Santé et de la Recherche Médicale, Univ. Lille, F-59000 Lille, France; 6Laboratoire Excellence Development of Innovative Strategies for a Transdisciplinary Approach to ALZ-heimer’s Disease (DISTAlz), Institut National de la Santé et de la Recherche Médicale, Univ. Lille, F-59000 Lille, France; 7Department of Emergency Psychiatry and Acute Care, Lapeyronie Hospital, Centre Hospitalier Régional Universitaire de Montpellier, F-34000 Montpellier, France; e-olie@chu-montpellier.fr; 8Pathologies du Systeme Nerveux: Recherché Épidémiologique et Clinique (PSNREC), Université de Montpellier, Institut National de la Santé et de la Recherche Médicale, Centre Hospitalier Régional Universitaire de Montpellier, F-34000 Montpellier, France; 9Centre Mémoire Ressources et Recherche (CMRR), Service de Neurologie, Centre Hospitalier Régional Universitaire de Besançon, F-25000 Besançon, France; eloi.magnin@laposte.net; 10Neurosciences Intégratives et Cliniques EA481, Université de Bourgogne Franche-Comté, F-25000 Besançon, France; djamila.bennabi@univ-fcomte.fr; 11Unité Cognitivo-Comportementale of Bailleul, Centre Mémoire Ressource et Recherche, University Lille Nord de France, F-59270 Bailleul, France; florence.lebert@epsm-fl.fr; 12Clinical and Research Memory Center of Montpellier, Department of Neurology, Gui de Chauliac Hospital, University of Montpellier, Institut National de la Santé et de la Recherche MédicaleU1061, F-34000 Montpellier, France; audrey.gabelle@chu-montpellier.fr; 13Psychiatry Center, Expert Center FondaMental Fondation, Clinical Research Center 1431, F-25030 Besançon, France; 14Day Hospital of Geriatrics, Memory Resource and Research Centre (CMRR) of Strasbourg, Department of Geriatrics, Hopitaux Universitaires de Strasbourg, F-67000 Strasbourg, France; frederic.blanc@chru-strasbourg.fr; 15University of Strasbourg and French National Centre for Scientific Research (CNRS), ICube Laboratory and Federation de Medecine Translationnelle de Strasbourg (FMTS), Team Imagerie Multimodale Integrative en Sante (IMIS)/ICONE, F-67000 Strasbourg, France; 16Université de Paris, Unité Mixte de Recherche 1144 Optimisation Thérapeutique en Neuropsychopharmacologie, Equipe Biomarqueurs de la Réponse Thérapeutique et de la Rechute en Neuropsychiatrie, Institut National de la Santé et de la Recherche Médicale, F-75010 Paris, France

**Keywords:** bipolar disorder, cognition, dementia, Alzheimer’s disease, biomarkers, Parkinsonism, neurodegenerescence

## Abstract

Bipolar disorder is associated with an increased risk of dementia with aging. Little is known regarding this association, limiting appropriate diagnosis and management. We aimed to describe the characteristics of bipolar patients with late cognitive impairment for whom the hypothesis of an underlying neurodegenerative disease had been raised. We performed a retrospective multicenter study, recruiting bipolar patients over 50 years old from five French tertiary memory centers who had undergone cerebrospinal fluid (CSF) biomarker assessment for Alzheimer’s disease (AD). Clinical, neuropsychological, and paraclinical characteristics were analyzed and 78 patients were included. The mean age at the onset of cognitive impairment was 62.4 years (±9.2). The mean MMSE score was 22.8 (±4.5), the mean FAB was 11.7 (±3.9), and the mean FCRST was 15.8 (±7.4)/36.8 (±9.7) (free/total recall). A total of 48.6% of the patients displayed cognitive fluctuations, and 38.2% showed cognitive improvement during follow-ups; and 56.3% of the patients showed Parkinsonism, of which 12.7% had never received antipsychotics. Among patients who underwent DAT-scans, 35.3% displayed dopaminergic denervation; 10.3% of patients had CSF AD biological signature (“A+ T+” profile), while 56.4% had other abnormal CSF profiles. Thus, clinical presentation was dominated by executive dysfunction, episodic memory impairment, fluctuating cognition, and a high frequency of Parkinsonism. Specifically, high frequency of delusional episodes suggests limited tolerance of psychotropic drugs. Most patients had abnormal CSF biomarker profiles, but only a minority displayed AD’s specific biomarker signature. Therefore, while our results unveil shared common neurocognitive features in bipolar patients with cognitive impairment of suspected neurodegenerative origin they suggest a participation of various underlying pathologies rather than a common degenerative mechanism in the pathophysiology of this condition.

## 1. Introduction

Patients with bipolar disorder (BD) display neuropsychological deficits in their attentional capacities, executive functions, and episodic verbal memory. These deficits may be detected as early as the first episode and persist throughout life, including periods of euthymia [1].

They are also at increased risk of dementia while aging [2], with an odds ratio ranging from 2.96 to 7.52 compared to the general population [3,4]. In contrast to early-onset cognitive deficits, late-onset disabling neurocognitive impairment has been underexplored, partly due to the absence of consensus definition and diagnostic criteria that have prevented detailed phenotypic description. As a consequence, pathophysiology of dementia that occur in patients with BD remains uncertain. A neurodegenerative process is most commonly suspected as some studies have shown the existence of accelerated brain aging in BD [5,6]. In addition, BD features the same abnormalities in several systems and signaling pathways as neurodegenerative diseases—i.e., Alzheimer’s disease (AD) and Lewy Body Dementia [7] and an association between BD and alpha-synucleinopathies (i.e., Parkinson’s disease and Lewy Body dementia) is increasingly recognized [8,9,10]. The main alternative hypothesis relies on neuroprogression, a concept postulating that the repetition of mood episodes would cause irreversible neuronal changes that can lead to increased relapse frequency, decreased response to treatment, and cognitive decline [11].

Poor understanding of the mechanisms underlying cognitive impairment of suspected neurodegenerative origin (CI-SNO) in patients with BD negatively impacts the management of these patients that are at risk of treatment adverse effects [12]. Advances in the field of biomarkers for neurodegenerative processes have dramatically modified the diagnosis and management of neurocognitive disorders. Indeed, it is now possible to detect in CSF—procured through a simple lumbar puncture—surrogate markers of AD’s pathological signature, such as beta amyloid and tau accumulation and neuronal death [13]. Consequently, diagnostic criteria have been updated for both clinical practice and research, so AD might now be diagnosed from the subjective cognitive impairment stage using the 2018 AT(N) criteria (A for amyloid, T for tau, and N for neurodegenerescence); an A+/T+ profile is associated with a high probability of having brain AD pathophysiology [14]. Unfortunately, these approaches have barely been used to explore patients with BD and CI-SNO. Indeed, only one study compared the values of CSF AD biomarkers between small groups of patients (14–25) with BD and mild cognitive impairment (BD-MCI), AD or amnestic mild cognitive impairment (aMCI) and controls. The authors found that biomarkers profiles of patients with BD-MCI were not different from those of patients with aMCI and controls and concluded that cognitive impairment in patients with BD was not associated with CSF AD biomarkers pathological signature [15].

In this study, we used the suspicion of an underlying neurodegenerative process that triggered CSF AD biomarkers assessment as an unbiased entry point to describe the clinical, neuropsychological, and paraclinical features of a cohort of patients with BD and disabling late cognitive impairment explored in five tertiary memory centers. We show that these patients, while heterogeneous, display unsuspected distinctive characteristics that should be taken into account for both clinical practice and further research.

## 2. Materials and Methods

### 2.1. Study Population

The inclusion criteria were: history of bipolar disorder diagnosed by a psychiatrist according to current diagnostic criteria at the time of diagnosis (DSM-III, IV, or V); age over 50 years at the time of first assessment; cognitive complaints with significant impact on daily functioning (according to managing clinician); and a detailed work-up, including (at least) neurological exam and lumbar puncture for cerebrospinal fluid (CSF) AD biomarker assessment. The exclusion criteria were: history of brain lesion that could affect neurological or neurocognitive examination (i.e., severe head trauma, ischemic or hemorrhagic stroke, intracranial tumor, and central nervous system infection, among others). Patients were restrospectively identified through a systematic sequential search in the medical records databases of five French tertiary memory centers (Paris, Lille, Montpellier, Strasbourg, and Besançon). Search workflow was: CSF AD biomarkers available > history of bipolar disorder > age over 50 years at first visit > at least one medical evaluation reporting neurological exam > no medical record or brain imaging indicative of brain lesion. All patients fulfilling the inclusion criteria and having no exclusion criteria were included.

### 2.2. Data Collection

The following data were extracted from patient’s medical records: demographics (age, gender, and education level); characteristics and evolution of BD; neurological examination and neuropsychological assessment; structural or metabolic brain imaging; and CSF AD biomarkers.

#### 2.2.1. History of Bipolar Disorder

The following data were collected regarding BD history: age at diagnosis; type of bipolar disorder (I or II) (when specified); alcohol use disorder; treatment history; and history of electroconvulsive therapy. Information that could be obtained only for a minority of patients (e.g., number of relapses, and number of hospitalizations) was not analyzed.

#### 2.2.2. Neurocognition and Evolution

The following data were collected regarding the history and clinical presentation of neurocognitive impairment: age at onset; age at first examination; and Parkinsonism, defined as the presence of at least one symptom among rest tremor, akinesia/bradykinesia and rigidity.

Regarding neuropsychological examination, results were collected from three tests assessing key dimensions of cognitive functioning: Mini-Mental State Examination [16] (MMSE, global cognitive efficiency); Frontal Assessment Battery [17] (FAB, executive functions); and Free and Cued Selective Reminding Test [18] (FCSRT, verbal episodic memory). Examinations had been performed locally by graduate neuropsychologists for clinical practice. Due to the retrospective design of the study, inter- and intra-rater reliability could not be assessed

Regarding the evolution of cognitive performance, the following data were collected: cognitive fluctuations over time; delusional episodes; and the difference between baseline MMSE and MMSE at the last follow-up (∆MMSE), with a minimum of three months between the two measures.

#### 2.2.3. Brain Structural and Metabolic Imaging

The following data from structural and functional brain imaging were retrieved: semi-quantitative assessment of hippocampal atrophy according to the Scheltens score (MRI) [19]; presence/absence of cortical hypometabolism (18-FDG PET); and presence/absence of dopaminergic denervation on DAT-scan. MRI results were retrieved if MRI protocol (performed for clinical practice) included at least axial T2 or T2-FLAIR, and coronal T2 or T1. Analyses of imaging parameters were provided by the specialist that performed the exam and confirmed by the doctor in charge of the management of the patient. Due to the retrospective design of the study, inter- and intra-rater reliability could not be assessed.

#### 2.2.4. CSF Alzheimer’s Disease Biomarkers

Quantification of CSF biomarkers levels were performed in each center using validated commercial sandwich ELISA kits. Levels of Aβ 1–42, Aβ 1–40, phosphorylated tau (tau-p), and total tau (tau-t) in the CSF were collected. Results were dichotomized according to local laboratory thresholds (Table 1) to determine the AT(N) status for each patient.

### 2.3. Descriptive Statistics

Quantitative variables were expressed as means, medians, and standard deviation. Qualitative variables were expressed as percentages.

## 3. Results

### 3.1. Bipolar Disorder

Data related to their psychiatric histories are summarized in Table 2. A total of 78 patients with BD met the inclusion criteria, including 29 men (37.2%), and 49 women (62.8%).

BD type was available in 32 patients (41%), with 21 patients having type I BD (26.9%) and 11 patients having type II BD (14.1%). Comorbid alcohol use disorder was present in 20 patients (25.6%). The mean age at diagnosis of BD was 42.5 years (standard deviation: 15.6; range: 14–75). Substratification found that 50.9% of patients were diagnosed before the age of 40, 13.3% between 40 and 50, 20.7% between 50 and 60, and 15.1% over 60 years of age.

Regarding management, 46 patients (59%) were on antipsychotics at the time of their initial assessment, 10 (12.8%) had a history of antipsychotic treatment, and 22 (28.2%) had never received long-term antipsychotics. Regarding lithium, 19 (27.5%) were receiving lithium at the time of first assessment, 8 (11.6%) had a history of lithium treatment, and 42 (60.9%) had never received lithium. Finally, 14.8% of patients had undergone electroconvulsive therapy sessions during the course of their disease.

### 3.2. Neurocognition

Neurological and neuropsychological data are summarized in Table 3. The average age at onset of cognitive impairment was 62.5 years (+/−9.2). The mean age at the first cognitive assessment was 67 years (+/−7.9). The average time before the first cognitive assessment was 4.6 years (+/−5.2). Cognitive fluctuations were reported in 34 patients (48.6%), while 21 patients (35%) had experienced delusional episodes.

Neurological examination revealed the presence of Parkinsonism in 40 patients (56.3%), of whom 27 (38%) were currently taking antipsychotics, 4 (5.6%) had a history of antipsychotic treatment, and 9 (12.7%) had never received long-term antipsychotic treatment.

The mean MMSE score was 22.8 (+/−4.5). Substratification on baseline MMSE found that 25 patients (32%) displayed mild cognitive impairment (MMSE > 25), while 35 patients (45%) had mild major neurocognitive impairment (20 < MMSE < 25), and 18 patients (23%) had moderate-to-severe major neurocognitive impairment (MMSE < 20).

The mean FAB score was 11.7 (+/−3.9). Regarding the FCSRT, the mean free recall score was 15.8 (+/−7.4) and the mean total recall score was 36.8 (+/−9.7).

Follow-up data were available for 55 patients. The mean follow-up time was 39.5 months (+/−50). Two cognitive progression profiles were observed: while 34 (61.8%) patients showed worsening cognitive performance over time (∆MMSE ≤ 0, with highest negative ∆MMSE = −20), 21 (38.2%) improved during follow-up (∆MMSE > 0, with highest positive ∆MMSE = 13).

### 3.3. Imaging and Biomarkers

Imaging and biomarker data are summarized in Table 4.

Brain MRI results were available for 40 patients. Among those, 16 patients (40%) displayed no hippocampal atrophy (Scheltens 0), 4 (10%) had incipient atrophy (Scheltens 1), 14 (35%) had moderate atrophy (Scheltens 2), and 6 (15%) showed marked atrophy (Scheltens 3). No patient displayed severe atrophy (Scheltens 4).

Results of dopamine transporter imaging (DAT-scan) and 18-FDG PET-CT imaging were available in 34 and 23 patients, respectively. DAT-scan results revealed significant dopaminergic denervation in the striatum in 35% of patients. In addition, 86.9% of patients who underwent 18-FDG PET-CT showed significant cortical hypometabolism, the topography of which was highly variable without a discernible common pattern.

CSF AD biomarker data were available for all patients. Eight patients (10.3%) displayed AD’s biological signature as defined by a A+/T+ biomarker profile. 44 patients (56.4%) presented an “intermediate” profile, with either an A+ (23 patients, 29.5%) or a T+ profile (21 patients, 26.9%). The remaining 26 patients (33.3%) had normal CSF biomarker results.

## 4. Discussion

In this study, we described the characteristics of a large cohort of patients with BD with late cognitive impairment of suspected neurodegenerative origin (CI-SNO). Despite significant heterogeneity in neurocognitive profiles, we could distinguish several remarkable features, such as Parkinsonism and/or abnormal dopamine transporter imaging, fluctuating cognition, and frequent improvement in cognition over time. We also found a limited frequency of CSF biomarkers that are indicative of AD’s pathophysiology.

While cognitive impairment in these patients was judged to be severe enough to raise concerns about an underlying neurodegenerative process, global cognitive performance was, on average, mildly impaired, as shown by the mean MMSE at first visit being 22.8. In addition, cognitive trajectories were characterized by marked fluctuations over time (including delusional episodes in one of three patients) and improvement in the global cognitive functioning during follow-ups in a large proportion of patients, with a gain of up to 13 points in the MMSE score. Several mechanisms may underlie this cognitive “lability”. First, cognitive impairment is a core symptom of depression and mania, and fluctuations in mood that occur through the course of BD cause variations in cognition [20]. A recent study performed in non-demented older patients with BD, however, identified that mood state only had a mild effect on cognitive status, which remained quite stable during a five-year follow-up period in this population [21]. Second, fluctuating cognition might be the “cognitive” counterpart of the increasing severity and recurrence of mood episodes over time, as postulated by the neuroprogression hypothesis [11]. Finally, psychotropic medications, especially those with anticholinergic properties, such as antipsychotics, might have negatively impacted cognition [22]. In our cohort, we observed a high frequency of antipsychotic use and low levels of lithium prescriptions. While a global decrease in lithium prescription has been well documented in recent studies [23], especially in older patients with BD [24], it is possible that the side effects of mood stabilizers might be amplified in patients with BD and CI-SNO.

It might be tempting to link the high prevalence of Parkinsonism observed in our cohort to the latter observation regarding antipsychotic drug prescription. 12.7% of patients with extrapyramidal symptoms, however, had no lifetime history of treatment with antipsychotics. In addition, a dopamine deficiency was observed in more than a third of patients with available DAT-scan data. Of note, similar findings have previously been described by Lebert and colleagues [25] and confirmed more recently using ^123^I-Ioflupane Dopamine Transporter SPECT [26]. In line with this, an increased risk for patients with BD to receive a diagnosis of Parkinson’s disease has been reported by several studies [8,9,27]. Moreover, Kohsravi recently reported the case of a patient with BD and CI-SNO that recapitulated most of Lewy Body’ Dementia features [10]. This patient showed very poor tolerance of various mood stabilizers—including lithium carbonate—that caused worsening of Parkinsonism. This observation suggested a complex interaction between neurodegenerative and iatrogenic features in the determinism of Parkinsonism observed in patients with BD and CI-SNO that had been previously emphasized by others [28]. Results of subgroup analyses performed in our cohort that found similar frequencies of Parkinsonism and antipsychotic use in patients with and without dopamine deficiency on DAT-Scan (Appendix A) support these conclusions.

Another argument for a possible underlying neurodegenerative process is the profile of episodic memory impairment displayed by our patients. Indeed, alteration of both free and cued recall corresponding to the so-called “amnestic hippocampal syndrome” is considered strongly suggestive of Alzheimer’s disease [29]. Yet, hippocampal atrophy, an archetypal MRI marker of AD [19], appeared limited in patients with BD and CI-SNO, with 85% of them displaying a Scheltens score ≤ 2. Moreover, we found that only eight patients (10.3%) had CSF AD biological signature (A+, T+) as compared to 29.6% A+, T+ profiles among 2000+ consecutive patients included in the Cognitive Neurology Center BioCogBank cohort (mean age at lumbar puncture = 71.3 (9.9), mean MMSE = 21.2 (5.8)); that cohort is representative of the tertiary memory centers that participated in the study. These results corroborate those previously obtained in a smaller population of elderly patients with BD with mild cognitive impairment [15], strongly suggesting that CI-SNO is not caused by definite AD disorder in the majority of these patients.

These observations do not exclude that some neurodegenerative changes might be involved in CI-SNO in patients with BD. Indeed, 56.4% of non-AD patients had abnormal CSF biomarker profiles, with either A+, T− or A−, T+ changes. According to current knowledge, “A+, T−” −referred to as “Alzheimer’s disease pathologic changes” in the 2018 criteria, reflects underlying brain amyloidopathy [30], which means extracellular deposits of aggregated amyloid beta peptides (amyloid plaques). According to the amyloid cascade hypothesis, these deposits are the initiators of AD’s pathophysiological process [31] and cognitively normal older A+ subjects show greater cognitive decline over time [32]. Importantly, brain amyloidopathy may also contribute to cognitive impairment in other neurodegenerative disorders, such as Lewy Body Dementia (LBD). Indeed, A+ LBD patients show an onset of cognitive impairment at an earlier age and more impaired visuo-spatial functions compared to patients without amyloidopathy (A−) [33]. Yet, A+ patients displayed similar profiles of hippocampal atrophy as A− patients in our cohort (Appendix A). A−, T+ profiles belong to the “non-Alzheimer’s disease pathologic change” group, also known as “Suspected Non-Alzheimer Pathology” (SNAP). The SNAP profile is identified in up to 22% of elderly healthy individuals and is not associated with significant cognitive decline over time in this population [34]. In a recent study, however, we showed that patients with cognitive impairment and a CSF A−, T+ profile mainly received a diagnosis of frontotemporal dementia (FTD) [35]. In addition, CSF p-tau levels were shown to correlate with cerebral tau pathology burden in patients with autopsy-confirmed frontotemporal lobar degeneration and AD [36].

Unfortunately, data regarding neuropathological findings in patients with BD and CI-SNO are scarce. One study only analyzed the neurodegenerative changes in the autopsy cases of 11 consecutive older patients with BD [37]. In this study, 6/11 patients fulfilled the diagnostic criteria of a definite neurodegenerative disease, mainly belonging to the “tauopathies” group. Moreover, all patients displayed argyrophilic grains, a type of tau lesions in neuronal processes, to some extent. Importantly, amyloid pathology was mild or absent in all patients included in this study.

Our approach based on clinical suspicion of a neurodegenerative process at stake allowed us to describe the largest cohort of patients with BD and late cognitive impairment affecting activities in daily living. However, it goes with its counterparts, i.e., lack of uniformity and missing data due to the retrieval of data from medical records; recruitment of subjects from tertiary memory clinics; lack of control group. Another limitation of our study resides in the limited data available regarding history of BD (i.e., BD type, frequency, polarity and severity of relapses, detailed treatment history, and so on), as the neuroprogression hypothesis postulates that there might be a lifetime cumulative effects from repeated brain damage caused by mood episodes. Another limitation is the absence of detailed objective evaluation of mood status at the time of cognitive evaluation. While memory centers’ good practice guidelines recommend not performing diagnostic neuropsychological testing in patients with incident depression and/or mania, we cannot rule out that some patients presented with mildly impaired mood at the time of evaluation. In addition, absence of consistency in imaging protocols (MRI, DAT-CT, PET/CT) prevented us from performing quantitative analyzes (i.e., volumetry, tracer uptake measurement, and so on). Finally, the relatively small number of patients and the retrospective and declarative design of the study may have been responsible for bias that could have prevented us from capturing the full spectrum of CI-SNO in patients with BD.

## 5. Conclusions

The results from this study plead against the idea that the CI-SNO observed in some patients with BD is mainly caused by a unique definite neurodegenerative disorder. Rather, they suggest that various neurodegenerative changes might act synergistically with other factors linked to the disease (e.g., mood status, medications, and neuroprogression) as stressors promoting cognitive disturbances and specific features unveiled by this study, such as Parkinsonism and fluctuating cognition. Additionally, these results pave the way to consensus diagnostic criteria for CI-SNO in patients with BD that are urgently needed in order to allow further research on this clinical and research challenge.

## Figures and Tables

**Table 1 jpm-11-01183-t001:** Alzheimer’s disease biomarkers thresholds for each center.

Center	Aβ42 (pg/mL)	Aβ40/Aβ42	Tau-p (pg/mL)	Tau-t (pg/mL)
Paris	860	12	22	225
Montpellier	500	10	60	400
Lille	700	15.4	60	400
Strasbourg	600	12	75	500
Besançon	700	15.6	60	450

Patients with an Aβ42 level below the cutoff, or with a Aβ40/Aβ42 ratio above the cutoff, were considered “A+”. Patients with a phosphorylated tau (Tau-p) or total tau (Tau-t) level above the cutoffs were considered “T+” or “N+”, respectively.

**Table 2 jpm-11-01183-t002:** Bipolar disorder history and treatment.

	Mean (s.d.)	Median
Age (years)	73.3 (8.6)	72
Age at diagnosis of bipolar disorder (years)	42.5 (15.6)	40
	Number (*n*)	Proportion (%)
Gender		
Male	29	37.2
Female	49	62.8
Bipolar disorder type		
Type 1	21	65.6
Type 2	11	34.4
Unknown	46	
Age at diagnosis		
Before 40 years	27	50.9
Between 40 and 50 years	7	13.3
Between 50 and 60 years	11	20.7
After 60 years	8	15.1
Unknown	25	
Alcohol use disorder		
Present	20	25.6
Absent	58	74.4
History of electroconvulsive therapy		
Present	9	14.8
Absent	52	85.2
Unknown	17	
Antipsychotics		
Ongoing	46	59
Past	10	12.8
Never	22	28.2
Lithium		
Ongoing	19	27.5
Past	8	11.6
Never	42	60.9
Unknown	9	

**Table 3 jpm-11-01183-t003:** Neurocognitive features.

	Mean (s.d.)	Median
Age at onset of cognitive impairment (years)	62.4 (9.2)	62
Age at first neurological assessment (years)	67.0 (7.9)	66.9
Time before first cognitive assessment (years)	4.6 (+/−5.2)	3.3
Follow-up time (moths)	39.5 (+/−50)	13.5
MMSE score at first assessment	22.8 (4.5)	24
Most recent MMSE score	23.1 (5.1)	24
∆MMSE	0 (5.3)	0
FCSRT score (free recall)	15.8 (7.4)	15
FCSRT score (total recall)	36.8 (9.7)	39
FAB score	11.7 (3.9)	11
	Number (*n*)	Proportion (%)
MMSE groups		
MMSE > 25	25	32
20 ≤ MMSE ≤ 25	35	45
MMSE < 20	18	23
∆MMSE evolution		
∆MMSE > 0	21	38.2
∆MMSE < 0	34	61.8
Unknown	23	
Cognitive fluctuations		
Present	34	48.6
Absent	36	51.4
Unknown	8	
Delusional episodes		
Present	21	35
Absent	39	65
Unknown	18	
Parkinsonism		
Under antipsychotics	27	38
History of antipsychotics	4	5.6
No antipsychotics (never in lifetime)	9	12.7
Not present	31	43.7
Unknown	7	

**Table 4 jpm-11-01183-t004:** Imaging and biomarkers.

	Number (*n*)	Proportion (%)
CSF AD biomarkers		
“A+T+” profile	8	10.3
“A+” profile	23	29.5
“T+” profile	21	26.9
Normal	26	33.3
Hippocampal atrophy		
Scheltens 0	16	40
Scheltens 1	4	10
Scheltens 2	14	35
Scheltens 3	6	15
Scheltens 4	0	0
Unknown	38	
Dopamine deficiency on DAT-CT		
Present	12	35.3
Absent	22	64.7
Unknown	44	
Cortical hypometabolism on TEP-CT 18-FDG		
Present	20	86.9
Absent	3	13.1
Unknown	55

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
