# Peer review of "Characteristics of Bipolar Patients with Cognitive Impairment of Suspected Neurodegenerative Origin: A Multicenter Cohort"

_jpm, 2021, doi:10.3390/jpm11111183_

Round 1
Reviewer 1 Report
The authors gave a broad overview of the clinical and paraclinical features in elderly patients with bipolar disorders with suspected neurodegeneration as underlying cause of a cognitive impairment. They showed the aspects of neurodegeneration in bipolar patients using retrospective data including neuropsychological assessment, hippocampal atrophy in the brain MRI, measuring dopaminic deficiency via dopamine transporter imaging and cortical hypometabolism in 18-FDG-PET-CT. While Alzheimer´s pathology usually plays a major role in elderly patients with cognitive decline without a previous history of psychiatric diseases, the spectrum of neurodegenerative causes in this study is more diverse.
These are my comments.
- The exact connection between mood and cognitive symptoms remains partially unclear yet, even there are some hints that cognition may be relatively stable during the disease course. Therefore an explanation is needed in which phase of the bipolar disorder the neuropsychological tests were performed.
- Parkinsonian symptoms were seen in more than half of the patients and a dopamine deficiency in one third of the patients which underwent a DAT-scan. Unfortunately the authors gave no information, whether the clinical impression of Parkinsonism correlated with the findings of the dopamine transporter imaging. Complicating DAT-scans may result pathological in various neurodegenerative diseases and can be influenced by medications, especially antidepressants (which were probably used in at least some of the patients in this study). Regarding the relatively small cohort I would suggest giving a more differentiated overview of the group of patients with Parkinsonism to distinguish between a parkinsonoid after treatment with neuroleptics and other non-treatment-associated forms of Parkinsonism.
- Moreover I would recommend mentioning the used criteria/definition/scale of parkinsonian symptoms/movement disorders in the section 2 (Material and Methods).
- Finally some information (if available) regarding the used neuroleptics (typical vs. atypical) in the above mentioned group with treatment associated Parkinsonism would be helpful.
- The authors analyzed brain MRI results (using the Scheltens score of hippocampal atrophy) and CSF data. Again there is no information available concerning a correlation of the CSF biomarkers and the MRI findings (e. g. regarding an underlying Alzheimer’s pathology). Moreover only using an atrophy scale of the hippocampus without any other specification of atrophy patterns seems shortsighted, especially in a group of patients with relatively early onset of cognitive symptoms (62 years).
- At last I would recommend adding the used MRI sequences in the section 2.2.3. (Brain structural and metabolic imaging).
Author Response
We thank our reviewer for his careful reading of our paper and his many interesting questions and suggestions. As a preliminary comment, we would like to emphasize that the condition we are interested in in this paper, i.e. cognitive impairment of suspected neurodegenerative origin in patients with bipolar disorder, has been barely explored because of several limitations, the most important being the absence of consensus diagnostic criteria and, as a consequence, the absence of prospective research cohort. Thus to conduct this study we adoped a pragmatic approach by defining the condition not by a set of clinical or paraclinical criteria, but by the clinical impression of the memory clinic expert at the time of first evaluation that led him to perform a lumbar puncture for AD CSF biomarkers assessement. This allowed us to build the first cohort of patients with BD and CI-SNO using clinical practice registries. Unfortunately, while this method provided us with abundant and relevant, unbiased data, its counterparts are linked to the essence of data retrieved from medical records that is to say lack of consistency in the initial work-up and follow-up between patients and missing data.
We elaborated on these points in the intro and limits sections of the manuscript.
- The exact connection between mood and cognitive symptoms remains partially unclear yet, even there are some hints that cognition may be relatively stable during the disease course. Therefore an explanation is needed in which phase of the bipolar disorder the neuropsychological tests were performed.
This is a very important point and a known limit of our study. We elaborated on this in the limits section of the manuscript.
- Parkinsonian symptoms were seen in more than half of the patients and a dopamine deficiency in one third of the patients which underwent a DAT-scan. Unfortunately the authors gave no information, whether the clinical impression of Parkinsonism correlated with the findings of the dopamine transporter imaging. Complicating DAT-scans may result pathological in various neurodegenerative diseases and can be influenced by medications, especially antidepressants (which were probably used in at least some of the patients in this study). Regarding the relatively small cohort I would suggest giving a more differentiated overview of the group of patients with Parkinsonism to distinguish between a parkinsonoid after treatment with neuroleptics and other non-treatment-associated forms of Parkinsonism.
As suggested by our reviewer, we performed a subgroup analysis among patients with and without dopamine deficiency regarding frequency of Parkinsonism and antipsychotics use. Interestingly, this analysis revealed no differences between the two groups. We added a table (supplementary table 1) and amended the manuscript to include these results : « … This observation suggested a complex interaction between neurodegenerative and iatrogenic features in the determinism of Parkinsonism observed in patients with BD and CI-SNO that had been previously emphasized by others. Results of subgroup analyses performed in our cohort that found similar frequencies of Parkinsonism and antipsychotic use in patients with and without dopamine deficiency on DAT-Scan (supplementary table 1) support these conclusions.»
- Moreover I would recommend mentioning the used criteria/definition/scale of parkinsonian symptoms/movement disorders in the section 2 (Material and Methods).
We amended Material and Methods section 2.2.3 to clarify this point : « The following data were collected regarding the history and clinical presentation of neurocognitive impairment: age at onset; age at first examination; and parkinsonism, defined as the presence of at least one symptom among rest tremor, akinesia/bradykinesia and rigidity. »
- Finally some information (if available) regarding the used neuroleptics (typical vs. atypical) in the above mentioned group with treatment associated Parkinsonism would be helpful.
This information would have been of much value. Unfortunatelly, as the number of patients is limited and data has been retrieved from heterogeneous clinical data, we chose a single criteria « antipsychotics » when we designed the worksheet for data retrieval at the beginning of our study. Thus, we cannot provide our reviewer with these informations.
- The authors analyzed brain MRI results (using the Scheltens score of hippocampal atrophy) and CSF data. Again there is no information available concerning a correlation of the CSF biomarkers and the MRI findings (e. g. regarding an underlying Alzheimer’s pathology). Moreover only using an atrophy scale of the hippocampus without any other specification of atrophy patterns seems shortsighted, especially in a group of patients with relatively early onset of cognitive symptoms (62 years).
As suggested we analysed hippocampal atrophy in the 40 patients with available quantification depending on the amyloïd status (A+ or A-) and found similar profiles of atrophy in both groups. We added a table (supplementary table 2) and amended the manuscript to include these results : « Yet, A+ patients displayed similar profiles of hippocampal atrophy as A- patients in our cohort (supplementary table 2). »
The comment on the limits of the semi-quantitative Scheltens scale to assess hippocampal atrophy is highly relevant. Unfortunately, as data have been retrieved from medical records, MRI protocol were not consistent among patients. For instance, 3DT1 (needed to measure hippocampal volume through automatic segmentation) was available only for a minority of patients. We added a note on this point in the limits section of the manuscript: « In addition, absence of consistency in imaging protocols (MRI, DAT-CT, PET/CT) pre-vented us from performing quantitative analyzes (i.e. volumetry, tracer uptake measure-ment…). »
At last I would recommend adding the used MRI sequences in the section 2.2.3. (Brain structural and metabolic imaging).
We clarified this point in the material and methods section 2.2.3 of the manuscript : « MRI results were retrieved if MRI protocol (performed for clinical practice) included at least axial T2 or T2-FLAIR, and coronal T2 or T1. »

Reviewer 2 Report
The authors aimed to describe the characteristics of BD patients with late cognitive impairment for whom the hypothesis of an underlying neurodegenerative disease had been raised. They recruited 78 BD patients over 50 years old from five French tertiary memory centers who had undergone CSF biomarker assessment for Alzheimer's disease. They also analyzed data of clinical features, neuropsychological performance, and structural and metabolic neuroimaging. However, this study was limited by several methodological shortcomings, such as small sample size, lack of control group, large proportions of the missing data in clinical features and neuroimaging, and unclear procedures of the sampling and examinations, and thus diminish enthusiasm for publishing this work as is.
My specific comments and questions are as follows:
- Why did authors mention neurodevelopmental abnormalities in the first paragraph of the Introduction? It is confusing since this study aimed to investigate neurodegeneration. Please revise this paragraph.
- In the fourth paragraph of the Introduction, the authors cited reference [17] in the last sentence. Please summarize findings of this reference in the Introduction. Also, please consider to put the third and fourth paragraph in the same paragraph.
- It is unclear what was the study hypothesis? Please elaborate this in the Introduction.
- In Materials and Methods 2.1. Study population, it is unclear what was the procedure to identify 78 patients. For instance, how many patients were treated at these five tertiary memory centers? Were these 78 patients the only subjects who met study criteria (or any other potential subjects not included)? Please clarify this.
- One of the inclusion criteria in this study was the detailed work-up with neurological examination, neuropsychological evaluation, and lumbar puncture. However, many subjects actually did not have detailed information on these examinations (as shown in Table 4). Please clarify this.
- Please provide procedures for neuropsychological examinations (e.g., Who did the examinations? Was there any training to qualify examiners? Was there any data of inter- or intra-rater reliability?).
- Please provide procedures for brain structural and metabolic imaging (e.g., What was the equipment? How to make imaging acquisition and analysis? Was there any data of inter- or intra-rater reliability?).
- Please provide methods used to quantify CSF biomarkers.
- In Results 3.1. Bipolar disorder, "BD type was available in 33 patients..." was not matched with data shown in Table 2.
- In Table 2 and 3, Please use "n" to represent case number rather than "N".
- As already done in Table 4, please conduct subgroup analysis of neurocognitive features in those patients with cognitive impairment using MMSE scores as a cut-off point. Also, please compare these neurocognitive features with those of patients without cognitive impairment.
- In Discussion, several limitations of this study, such as small sample size, lack of control group, recruitment of subjects from tertiary clinics, etc., were not mentioned. Please provide discussions.
- It is unclear whether this study was approved by IRB? Please provide detailed information regarding the IRB approval (e.g., full name of IRB, approval number) and inform consent procedures.
- Please use abbreviation consistently. For instance, while some sentences use "patients with bipolar disorder" in the article, others use "patients with BD". In addition, please use "patients with bipolar disorder" instead of "bipolar patients".
- English should be editing by a professional English editor.
Author Response
We thank our reviewer for his careful reading of our paper and his many interesting questions and suggestions. As a preliminary comment, we would like to emphasize that the condition we are interested in in this paper, i.e. cognitive impairment of suspected neurodegenerative origin in patients with bipolar disorder, has been barely explored because of several limitations, the most important being the absence of consensus diagnostic criteria and, as a consequence, the absence of prospective research cohort. Thus to conduct this study we adoped a pragmatic approach by defining the condition not by a set of clinical or paraclinical criteria, but by the clinical impression of the memory clinic expert at the time of first evaluation that led him to perform a lumbar puncture for AD CSF biomarkers assessement. This allowed us to build the first cohort of patients with BD and CI-SNO using clinical practice registries. Unfortunately, while this method provided us with abundant and relevant, unbiased data, its counterparts are linked to the essence of data retrieved from medical records that is to say lack of consistency in the initial work-up and follow-up between patients and missing data.
We elaborated on these points in the intro and limits sections of the manuscript.
- Why did authors mention neurodevelopmental abnormalities in the first paragraph of the Introduction? It is confusing since this study aimed to investigate neurodegeneration. Please revise this paragraph.
We understand our reviewer’s confusion and amended the paragraph by suppressing the sentence : « As similar cognitive impairments have been detected in healthy first-degree relatives, it has been suggested that these deficits might be – at least partially – due to neurodevelopmental abnormalities. »
- In the fourth paragraph of the Introduction, the authors cited reference [17] in the last sentence. Please summarize findings of this reference in the Introduction. Also, please consider to put the third and fourth paragraph in the same paragraph.
We amended the introduction in accordance with these and summarized the findings of Forlenza et al. “Indeed only one study compared the values of CSF AD biomarkers between small groups of patients (14-25) with BD and mild cognitive impairment (BD-MCI), AD or amnestic mild cognitive impairment (aMCI) and controls. The authors found that biomarkers pro-files of patients with BD-MCI were not different from those of patients with aMCI and con-trols and concluded that cognitive impairment in patients with BD was not associated with CSF AD biomarkers pathological signature [15].”
- It is unclear what was the study hypothesis? Please elaborate this in the Introduction.
We elaborated on this point in paragraph 2 and 4 to better emphasize the working hypothesis (using CSF AD biomarkers assessment as a surrogate marker of suspicion of neurodegenerative origin as an unbiased entry point to build a cohort of patients with BD and late disabling impairment) that makes the originality, the relevance and the strength of this study. : « In this study, we used the suspicion of an underlying neurodegenerative process that triggered CSF AD biomarkers assessment as an unbiased entry point to describe the clini-cal, neuropsychological, and paraclinical features of a cohort of patients with BD and disabling late cognitive impairment explored in five tertiary memory centers. »
- In Materials and Methods 2.1. Study population, it is unclear what was the procedure to identify 78 patients. For instance, how many patients were treated at these five tertiary memory centers? Were these 78 patients the only subjects who met study criteria (or any other potential subjects not included)? Please clarify this.
Material and methods section 2.1 was amended to provide reader with more details regarding the inclusion procedure : « Patients were restrospectively identified through a systematic sequential search in the medical records databases of five French tertiary memory centers (Paris, Lille, Montpellier, Strasbourg, and Besançon). Search workflow was: CSF AD biomarkers available > history of bipolar disorder > age over 50 years at first visit > at least one medical evaluation re-porting neurological exam > no medical record or brain imaging indicative of brain lesion. All patients fulfilling the inclusion criteria and having no exclusion criteria were included.»
Unfortunately, as data were extracted from medical records databases, total number of patients and starting date of the database could not be retrieved.
- One of the inclusion criteria in this study was the detailed work-up with neurological examination, neuropsychological evaluation, and lumbar puncture. However, many subjects actually did not have detailed information on these examinations (as shown in Table 4). Please clarify this.
Answer to point 4 allowed us to clarify inclusion criteria. As shown in Table 4, 78/78 patients had available CSF AD biomarkers results. We extracted only information regarding Parkinsonism from neurological exam data. Patients were classified as unknown regarding this criterion if Parkinsonism was not mentioned in the medical record (neither as present or absent). Availability of detailed imaging data was not used as an inclusion criterion and brain imaging was thus not available for all patients.
- Please provide procedures for neuropsychological examinations (e.g., Who did the examinations? Was there any training to qualify examiners? Was there any data of inter- or intra-rater reliability?).
We clarified this point in the section 2.2.2 of the manuscript: « Examinations had been performed locally by graduate neuropsychologists for clinical practice. Due to the retrospective design of the study, inter and intra-rater reliability could not be assessed »
- Please provide procedures for brain structural and metabolic imaging (e.g., What was the equipment? How to make imaging acquisition and analysis? Was there any data of inter- or intra-rater reliability?).
We clarified this point in the 2.23. of the manuscript: « MRI results were retrieved if MRI protocol (performed for clinical practice) included at least axial T2 or T2-FLAIR, and coronal T2 or T1. Analyses of imaging parameters were provided by the specialist that performed the exam and confirmed by the doctor in charge of the management of the patient. Due to the retrospective design of the study, inter and intra-rater reliability could not be assessed. »
- Please provide methods used to quantify CSF biomarkers.
We clarified this point in the 2.2.4 section of the manuscript: « Quantification of CSF biomarkers levels were performed in each center using validated commercial sandwich ELISA kits. »
- In Results 3.1. Bipolar disorder, "BD type was available in 33 patients..." was not matched with data shown in Table 2.
We thank our reviewer for his careful reading. After double-checking data source we reconciled these numbers (typo in the text).
- In Table 2 and 3, Please use "n" to represent case number rather than "N".
This typo was corrected
- As already done in Table 4, please conduct subgroup analysis of neurocognitive features in those patients with cognitive impairment using MMSE scores as a cut-off point. Also, please compare these neurocognitive features with those of patients without cognitive impairment.
We thank our reviewer for this suggestion and we acknowledge that it is a very interesting question to explore. We performed the analyses. Their main results are shown in the table below. We think however that the meaning of these results are debatable and we wonder if they should be added to the manuscript. Indeed, defining relevant MMSE cut-offs that is always a difficult matter looks even more uncertain in the context of our study that intend to describe the characteristics of a poorly known condition. In addition, significance of baseline MMSE in this context of highly fluctuating cognition and frequent improvement over time looks uncertain.
We would be happy to get our reviewer and editor’s opinion on this point and we will of course add this table to the supplementary data if they think it is worth doing so.
|
|
MMSE ≤ 20 |
MMSE 21-25 |
MMSE > 25 |
|
|
|
Mean (s.d.) |
Mean (s.d.) |
Mean (s.d.) |
p |
|
Age at onset of cognitive impairment |
64.9 (7.2) |
62 (10.9) |
62.6 (9.8) |
0.463 |
|
Age at diagnosis of BD |
39.2 (16) |
46.2 (18) |
42.1 (14.2) |
0.490 |
|
|
Proportion |
Proportion |
Proportion |
|
|
Parkinsonism |
55 |
66.7 |
47.8 |
0.421 |
|
Fluctuating cognition |
55.6 |
56.5 |
36 |
0.284 |
|
CSF AD biomarkers |
|
|
|
|
|
"A+T+" profile |
13.6 |
4 |
16 |
0.363 |
|
"A+" profile |
45.5 |
24 |
24 |
0.191 |
|
"T+" profile |
27.2 |
20 |
40 |
0.290 |
|
Any abnormal profile |
59.1 |
44 |
48 |
0.57 |
|
Hippocampal atrophy |
|
|
|
0.588 |
|
Scheltens 0 |
33.3 |
37.5 |
42.9 |
|
|
Scheltens 1 |
0 |
6.2 |
21.43 |
|
|
Scheltens 2 |
44.4 |
43.8 |
21.43 |
|
|
Scheltens 3 |
22.2 |
12.5 |
14.3 |
|
|
Scheltens 4 |
0 |
0 |
0 |
|
|
Dopamine deficiency on DAT-CT |
25 |
57.1 |
10 |
0.046 |
- In Discussion, several limitations of this study, such as small sample size, lack of control group, recruitment of subjects from tertiary clinics, etc., were not mentioned. Please provide discussions.
We elaborated on these limitations in the limit section of the discussion: « Our approach based on clinical suspicion of a neurodegenerative process at stake al-lowed us to describe the largest cohort of patients with BD and late cognitive impairment affecting activities in daily living. However, it goes with its counterparts i.e. lack of uni-formity and missing data due to the retrieval of data from medical records; recruitment of subjects from tertiary memory clinics; lack of control group. »
- It is unclear whether this study was approved by IRB? Please provide detailed information regarding the IRB approval (e.g., full name of IRB, approval number) and inform consent procedures.
As data were anonymized, retrieved directly from medical records and no research procedure were performed in these patients, neither informed consent nor IRB approval were needed for this study.
- Please use abbreviation consistently. For instance, while some sentences use "patients with bipolar disorder" in the article, others use "patients with BD". In addition, please use "patients with bipolar disorder" instead of "bipolar patients".
We reconciled abbreviations and formulations throughout the manuscript.
- English should be editing by a professional English editor.
Manuscript had been proofread by a native english speaker before submission. Yet, we acknowledge that the many amendments we made to respond to our reviewers’ comments might have introduced several mistakes. Unfortunately, we were not offered enough time to perform both revision and English edition, but we would be happy to do so if our manuscript is accepted for publication.

Reviewer 3 Report
I would like to thank you for inviting me to review this manuscript. The present cohort study aimed to assess the characteristics of bipolar patients with cognitive impairment of suspected neurodegenerative origin. I sincerely appreciate the work of the authors, as it raises an important and challenging area for investigation among patients with bipolar disorders. Overall, the paper is well structured and reads well. In my view, although the research idea is highly interesting, the study could be more precise. The following comments would be helpful to enhance the quality of the study to be endorsed by the scientific community.
In the discussion section, it is necessary to pay more attention to the pathophysiology of cognitive and motor disorders in patients with bipolar disorder. This can help the reader better understand the possible causes of movement problems in the absence of antipsychotics administration (please see Dols and Lemstra 2020, PMID: 31954093; Khosravi 2021, PMID: 33834565). In addition, a recent case report has provided evidence of exacerbation of Parkinsonism following lithium therapy in the comorbidity of bipolar I disorder and Lewy body dementia, which may be useful for future research (please see Khosravi 2021, PMID: 33834565).
Author Response
We thank our reviewer for his very encouraging comments. Deciphering the mechanisms underlying parkinsonism in patients with BD and CI-SNO is an exciting and complex challenge and the references proposed by the reviewer give very good insights on this topic. We extended our discussion and elaborated on this question and we included the suggested references and the supplementary analyses proposed by reviewer 1 that support the idea of a complex pathomechanism underlying Parkinsonism in our patients: « Moreover, Kohsravi recently reported the case of a patient with BD and CI-SNO that reca-pitulated most of Lewy Body’s Disease features [10]. This patient showed very poor toler-ance of various mood stabilizers - including lithium carbonate- that caused worsening of Parkinsonism. This observation suggested a complex interaction between neurodegenera-tive and iatrogenic features in the determinism of Parkinsonism observed in patients with BD and CI-SNO that had been previously emphasized by others [28]. Results of subgroup analyses performed in our cohort that found similar frequencies of Parkinsonism and an-tipsychotic use in patients with and without dopamine deficiency on DAT-Scan (supple-mentary table 1) support these conclusions.»

Round 2
Reviewer 1 Report
The authors addressed all of my comments. As they mentionend in their conclusion, there is a lot of further research needed to understand the complex mechanisms of cognitive decline in patients with (neuro-)psychiatric disorders with suspected neurodegeneration.
Reviewer 2 Report
I understand that a waiver of informed consent procedure can be granted due to the de-identified and retrospective nature of the data. However, the major concern of this study is not being approved and supervised by the IRB. The decisions on exemption are made by the IRB representative, not by the investigators themselves. This procedure is to assure that appropriate steps are taken to protect the rights and welfare of humans subjects in this study.